# REALISTIC-GESTURE: CO-SPEECH GESTURE VIDEO GENERATION THROUGH CONTEXT-AWARE GESTURE REPRESENTATION

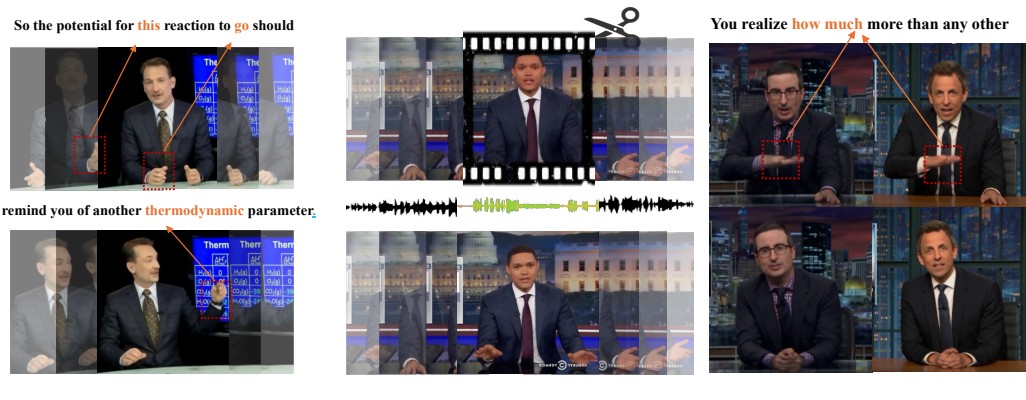

Long Sequence Generation  ·  Video Gesture Editing  ·  Gesture Pattern Transfer

Figure 1: Realistic-Gesture achieves various fine-grained control over video-level gesture motion.

## ABSTRACT

Co-speech gesture generation is crucial for creating lifelike avatars and enhancing human-computer interactions by synchronizing gestures with speech in computer vision. Despite recent advancements, existing methods often struggle with accurately aligning gesture motions with speech signals and achieving pixel-level realism. To address these challenges, we introduce Realistic-Gesture, a groundbreaking framework that transforms co-speech gesture video generation through three innovative components: (1) a speech-aware gesture tokenization that incorporate speech context into motion pattern representation, (2) a mask gesture generator that learns to map audio signals to gestures by predicting masked motion tokens, enabling bidirectional contextually relevant gesture synthesis and editing, and (3) a structure-aware refinement module that employs differentiable edge connection to link gesture keypoints to improve video generation. Our extensive experiments demonstrate that Realistic-Gesture not only produces highly realistic and speech-aligned gesture videos but also supports long-sequence generation and video gesture editing applications.

## 1 INTRODUCTION

In human communication, speech is often accompanied by gestures that enhance understanding and convey emotions De Ruiter et al. (2012). As these non-verbal cues play a vital role in effective interaction Burgoon et al. (1990), gesture generation a key component of natural human-computer interactions. As artificial intelligence advances, equipping virtual avatars with realistic gesture capabilities will become essential in creating immersive interactive experiences.

The relationships between the semantic and emotion content of speech context, the corresponding gestures, and the visual appearance of the speaker's performance are complex. As such, many

recent works Yi et al. (2023); Liu et al. (2023; 2022d;a) address a reduced form of this problem by generating a simplified representation of the 3D motion, consisting of joints and body parts, that plausibly accompanies a given speech sample, which can then be rendered using standard rendering pipelines. Such representations capture basic motion patterns, yet they neglect the importance of the speaker's visual appearance, resulting in a lack of realism that hinders effective communication.

Other works, *e.g.* ANGIE Liu et al. (2022c) and S2G-diffusion He et al. (2024), employ image-warping techniques, constrained by keypoints obtained from optical-flow-based deformations, for co-speech video generation. However, such approaches encounter several critical issues. First, these keypoints only define large-scale transformations, and thus miss subtle movements of specific body parts (*e.g.* hands, fingers). Second, it is difficult to connect such broad and unconstrained motions representation to speech content. This makes it difficult to conditionally generate gestures that are appropriately responsive to the audio, which inhibits the gestures' naturalism and expressivity. Finally, the generated motion patterns are often unstructured and overly reactive to large motion, resulting in noisy and imprecise renderings, especially in the hands and shoulders. Collectively, these challenges significantly limit the overall quality and realism of the generated video content.

To address these challenges, we introduce *Realistic-Gesture*, a framework designed to generate speech-aligned gesture motions and high-fidelity speech video outputs. Our approach begins with refined gesture motion representations using keypoints from pretrained human pose estimators, allowing for clearer disentanglement of human motions across the face, body, and hands. To uncover the intrinsic temporal connections between gestures and speech, we employ contrastive learning to align these two modalities. This joint representation captures the triggers of gesture patterns influenced by speech. We incorporate speech-contextual features into the tokenization process of gesture motions through knowledge distillation, aiming to infuse the gesture representations with implicit intentions conveyed in the audio. This integration creates a clear linkage between the gestures and the corresponding speech, enabling the conditional generation of gestures that accurately reflect the speaker's intended meaning based on the speech input. For latent motion generation, inspired by Muse Chang et al. (2023) and MAGE Li et al. (2023), we introduce a masked gesture generator that refines the alignment of gesture motions with the speech signal through bidirectional mask pretraining, enabling long sequence generation and editing capabilities. Finally, for uplifting the latent motion generation into 2D animations, we propose a structure-aware image refinement module that generates heatmaps of edge connections from keypoints, providing image-level supervision to improve the quality of body regions with large motion. Extensive experiments demonstrate that our method outperforms the existing state-of-the-art approaches in both quantitative and qualitative metrics.

In summary, our primary contributions are:

1. a *speech-aware gesture motion representation* obtained through knowledge distillation from the gesture-speech aligned features from contrastive learning;

2. a *masked gesture motion generator*, carefully designed to enable high-quality gesture motion generation with long sequence generation and edit-ability support; and

3. a *pixel-level refinement module*, which uses a structure-aware edge heatmap as supervision to improve the final output fidelity.

## 2 RELATED WORK

**Co-speech Gesture generation** Most recent works on co-speech gesture generation employ skeleton- or joint-level pose representations. Ginosar et al. (2019) use an adversarial framework to predict hand and arm poses from audio, and leverage conditional generation Chan et al. (2019) based on pix2pixHD Wang et al. (2018) for videos. Some recent works Liu et al. (2022d); Deichler et al. (2023); Xu et al. (2023) learns the hierarchical semantics or leverage contrastive learning to obtain joint audio-gesture embeding to assist the gesture pose generation. Rhythmic gesticulator Ao et al. (2022) construct high and low level audio-motion embedding based on lingustic theory for gesture generation. TalkShow Yi et al. (2023) estimates SMPL Pavlakos et al. (2019) poses, and models the body and hand motions for talk-show videos. CaMN Liu et al. (2022b) and EMAGE Liu et al. (2023) use large conversational and speech datasets for joint face and body modeling with diverse style control. ANGIE Liu et al. (2022c) uses unsupervised 2D keypoints with image-warping features based on MRAA Siarohin et al. (2021) to model body motion. It leverages

Vector Quantization van den Oord et al. (2018) to obtain common patterns, followed by a GPT-like network that outputs co-speech gesture videos. S2G-Diffusion He et al. (2024) uses TPS Zhao & Zhang (2022) and optical flow prediction to extract and refine latent motion features from videos. However, none of these works produce structure- and speech-aware motion patterns that are suitable for achieving natural and realistic gesture rendering.

**Conditional Video Generation**   Conditional Video Generation has undergo significant progress for various modalities, like text Blattmann et al. (2023), pose Karras et al. (2023); Wang et al. (2023b), and audio Ruan et al. (2023). Diffusion Models Ho et al. (2020) improve generation qualities. AnimateDiff Guo et al. (2024) presents an efficient motion adaptation module based on low-rank adaptation Hu et al. (2022) (LoRA) to adapt image diffusion model for video motion generation. AnimateAnyone Hu et al. (2023) construct referencenet for fine-grained control based on skeleton. Make-Your-Anchor Huang et al. (2024) improves avatar video generation through disentangled face and body based on SMPL-X conditions. Champ Zhu et al. (2024) introduces human SMPL models for guidance. EMO Tian et al. (2024) leverages audio as control signal for talking head generation. However, these methods are based on large amount of training data and slow in inference speed. None of them focus on the speech-gesture pixel-level video generation.

**Masked Representation Learning for Generation**   Masked Representation Learning has been demonstated an effective representation learning for various modalities. Devlin (2018); He et al. (2022) Some works explored the generation capabilities using this paradigm. MAGE Li et al. (2023) achieves high-quality image generation through iterative remasking. Muse Chang et al. (2023) extends this idea to leverage language with region masking for image editing and achieve fine-grained control. Recent Masking Models Pinyoanuntapong et al. (2024); Wang (2023); Mao et al. (2024) bring this strategy to the motion and gesture domain and improves the motion generation speed, quality, and editing capability. Inspired by these work, we propose the masked gesture generation conditioned the audio to learn the gesture-speech correspondence during generation.

## 3 PRELIMINARY

**Warping-Based Image Animation.**   Warping-based image animation methods have risen to prominence recently Siarohin et al. (2021; 2019); Zhao & Zhang (2022). They leverage keypoint predictor to identify pairwise corresponding keypoints between a source image and a driving image. This information is used to warp the source image to match the driving image, thereby producing a deformation that aligns with the driving scene. Following this, pixel-level optical flow and occlusion masks are estimated from the deformed images to capture global motion and handle occlusions for achieving driving image reconstruction. We defer additional details to the Appendix.

**Image-Animation Based Co-Speech Gesture Video Synthesis.**   In the context of co-speech gesture video synthesis, recent advancements have employed warping-based image animation techniques to derive motion patterns and learn the correspondence between these patterns and audio, facilitating speech-driven generation. Given a video clip $V = \{I_0, I_1, \ldots, I_N\}$ and an accompanying audio sequence $A = \{a_1, a_2, \ldots, a_N\}$, the objective is to predict motion representations $\hat{M}$ based on the initial frame $I_0$ and the audio input. The image animation module reconstructs all video frames $\hat{I}_1$ through unsupervised learning to derive motion representations and transformations. The audio sequence serves as guidance for reconstructing motion patterns across the entire sequence of frames following the initial frame.

However, this approach faces three significant challenges: (1) keypoints derived from global optical-flow-based transformations, learned through unsupervised methods, often fail to capture subtle movements of specific body parts; (2) the motion representations do not include contextual information from the speech, making it difficult to generate gestures that are conditionally responsive to audio; and (3) the lack of structural awareness in the motion representations leads to blurry and noisy predictions, particularly affecting the hands and shoulders, while also rendering the system sensitive to large motion patterns. To address these challenges, we propose the following methods to enhance control over co-speech gesture video generation.

## 4 REALISTIC-GESTURE

As shown in Fig. 2, our framework targets at generating realistic gesture videos. To achieve this goal, we first learn gesture-speech alignment to build speech-aware gesture motion representation through contrastive learning. (Sec. 4.1) To achieve fine-grained control over the gesture motion generation, we propose a Masking-based Gesture Generator, with long sequence and editing capabilities. (Sec. 4.2). To improve the noisy hand and shoulder movement during the uplifting of latent motion to pixel space, we propose a structure-aware image refinement through differentiable edge heatmaps for guidance. (Sec. 4.3).

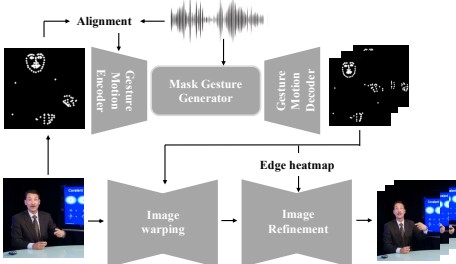

Figure 2: An overview of our framework.

### 4.1 SPEECH-AWARE GESTURE MOTION REPRESENTATION

Unlike ANGIE Liu et al. (2022c) and S2G-Diffusion He et al. (2024), which rely on unsupervised learning of keypoints for warping-based on optical flow, we utilize 2D poses extracted from images. While using 2D poses for image warping may slightly decrease fidelity, it significantly enhances the perceptual quality of the generated video, shown in Sec. 5.4. With poses, gestures can be decomposed into facial and body movements. We represent a gesture motion sequence as $G = [F; B] = [f_t; b_t]_{t=1}^{T}$, where $T$ denotes the length of the motion, $f$ represents the 2D facial landmarks, and $b$ denotes the 2D body landmarks. Further details on gesture representations can be found in the appendix. For speech representation, we extract audio embeddings from WavLM Chen et al. (2022). In addition, we extract Mel spectrogram features Rabiner & Schafer (2010) and beat information using librosa McFee et al. (2015). These features are concatenated to form the speech representation. For image-warping transformation, we select TPS Zhao & Zhang (2022).

**Speech-Gesture Alignment.** To align gesture motion patterns with the content of speech and beats, we draw inspiration from image-language contrastive learning Radford et al. (2021). We first project both speech and gesture modalities into a shared embedding space to enhance the speech content awareness of gesture features. As illustrated in Fig. 3, we separately train two gesture content encoders, $\mathcal{E}_f$ for face motion and $\mathcal{E}_b$ for body motion, alongside two speech encoders, $\mathcal{E}_{S_f}$ and $\mathcal{E}_{S_b}$, to map face and body movements and speech signals into this joint embedding space. For simplicity, we represent the general gesture motion sequence as $G$. We then apply mean pooling to aggregate content-relevant information from each feature sequence, resulting in the embeddings $z^s$ and $z^g$ for speech and gestures, respectively. We leverage CLIP-style contrastive learning to train these content encoders. Given a batch of paired embeddings $\mathcal{B} = \{(z_i^t, z_i^g)\}_{i=1}^{B}$, we optimize the following loss with $\tau$ as the temperature:

$$\mathcal{L}_{\text{contrast}} = -\frac{1}{B} \sum_{i=1}^{B} \left( \log \frac{\exp(z_i^g \cdot z_i^t / \tau)}{\sum_{j=1}^{B} \exp(z_i^g \cdot z_j^t / \tau)} + \log \frac{\exp(z_i^t \cdot z_i^g / \tau)}{\sum_{j=1}^{B} \exp(z_i^t \cdot z_j^g / \tau)} \right) \quad (1)$$

Unlike previous methods Ao et al. (2022); Liu et al. (2022d); Deichler et al. (2023), which primarily capture sequence-level alignment and may overlook local temporal dynamics, to mitigate this limitation, we randomly mask 30% of segments from both speech and gesture sequences within the same temporal regions during training. Furthermore, we apply a linear classifier on the gesture embedding to predict speech beats, enhancing the temporal alignment between gestures and speech. We defer additional details of temporal-level improvement by our strategy in the Appendix.

**Speech-Pattern Learning Through Knowledge Distillation.** For gesture motion tokenization, we utilize Residual Vector Quantization Lee et al. (2022) (RVQ) to capture the high diversity and complexity of facial and body motions. To construct context-aware motion representations, we directly encode alignment information into the gesture motion codebook. This allows the semantics and contextual triggers from speech (e.g., pronouns like "this" or "they") to be fused into the motion embedding, and enables the generator to easily identify the corresponding motion representation in response to speech triggers. To achieve this goal, we leverage gesture content encoder as the teacher and distill knowledge to codebook latent representation. We aim to maximize the cosine similarity

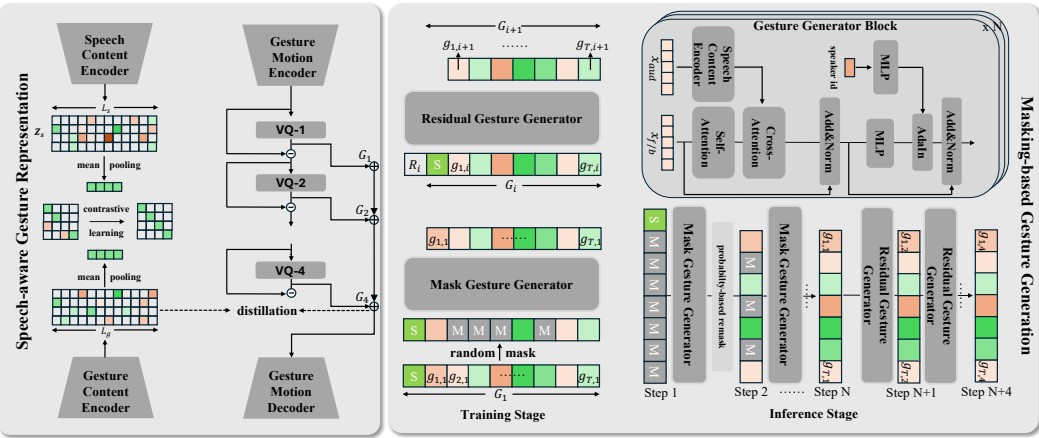

Figure 3: Left: Contrastive Learning for gesture-speech alignment. We distill the joint speech contextual-aware feature into latent codebook. Right: We use speech for generating gesture motion tokens with Mask Gesture Generator. We apply random mask for reconstruction during training and iterative remask based on probability for inference. Residual Gesture Generator based on the base VQ-tokens to predict the residuals.

over time between the final RVQ quantization output and the representation from the gesture content encoder, formulated as follows:

$$\mathcal{L}_{\text{distill}} = \sum_{t=1}^{T} \cos\left(p(Q_R)^t, Es(G)^t\right) \tag{2}$$

where $p$ denotes a linear projection layer, $Q_R$ is the final quantization output from the RVQ-VAE, $Es(G)$ represents the output from the gesture content encoder, and $T$ is the total time frames. The overall training objective for the RVQ-VAE is defined as:

$$\mathcal{L}_{\text{rvq}} = \mathbb{E}_{x \sim p(x)}\left[\|x - \hat{x}\|^2\right] + \alpha \sum_{r=1}^{R} \mathbb{E}_{z_r \sim q(z_r|x)}\left[\|e_r - \text{sg}\left(z_r - e_r\right)\|^2\right] + \beta\mathcal{L}_{\text{distill}} \tag{3}$$

where $\mathcal{L}_{\text{rvq}}$ combines a motion reconstruction loss, a commitment loss van den Oord et al. (2018) for each layer of quantizer with a distillation loss, with $\alpha$ and $\beta$ weighting the contributions.

## 4.2 SPEECH-CONDITIONED GESTURE MOTION GENERATION

To enhance the generation of gesture motions across different layers of the quantized codebooks, we draw inspiration from VALL-E Wang et al. (2023a) to design **Masked Gesture Generator** for jointly decoding facial and body motions for the base-layer outputs of quantizers, and **Residual Gesture Generator** for the face and body tokens from the subsequent $R$ residual quantization layers.

**Masked Gesture Generator.** As shown in Fig. 3, during training, we derive motion tokens by processing raw gesture sequences through both body and face tokenizers. The motion token corresponding to the source image acts as the conditioning for all subsequent frames. For speech control, we initialize the audio content encoder from alignment pre-training as described in Sec. 4.1. This pre-alignment of gesture tokens with audio encoder features enhances the coherence of gesture generation. We employ cross-attention, using the audio input as keys and values while the gesture representation serves as the query, integrating audio information with gesture feature. To refine control over gesture patterns, we apply Adaptive Instance Normalization (AdaIN) Huang & Belongie (2017) after the feed-forward layers, enabling diverse gesture styles based on the speaker's identity.

**Residual Gesture Generator.** The Residual Gesture Generator shares a similar architecture with the Masked Gesture Generator, but it includes $R$ separate embedding layers corresponding to each RVQ residual layer. During training, we randomly select a quantizer layer $j \in [1, R]$ for learning. All tokens from the preceding layers $t^{0:j-1}$ are embedded and summed to form the token embedding input. After generating the base layer predictions of discrete tokens from the Masked Gesture Generator, these tokens are fed into the Residual Gesture Generator. This module iteratively predicts the tokens from the base layers, ultimately producing the final quantized output.

**Inference.** While existing works Liu et al. (2023); Yi et al. (2023); Chen et al. (2024) leverage auto-regressive next-token prediction or diffusion-based generation process, these strategies hinder the fast synthesis for real-time applications. To resolve this problem, as in Fig. 3, we employ an iterative mask prediction strategy to decode motion tokens during inference. Initially, all tokens are masked except for the first token from the source frame. Conditioned on the audio input, the Mask Gesture Generator predicts probabilities for the masked tokens. In the $l$-th iteration, the tokens with the lowest confidence are re-masked, while the remaining tokens stay unchanged for subsequent iterations. This updated sequence continues to inform predictions until the final iteration, when the base-layer tokens are fully generated. Upon completion, the Residual Gesture Generator uses the predicted base-layer tokens to progressively generate sequences for the remaining quantization layers. Finally, all tokens are transformed back into motion sequences via the RVQ-VAE decoder.

**Training Objective.** To train our gesture generation models, $\mathcal{L}_{mask}$, and $\mathcal{L}_{res}$ functions for two generaors respectively by minimizing the categorical cross-entropy loss, as illustrated below:

$$\mathcal{L}_{mask} = \sum_{i=1}^{T} -\log p_\phi(t_i|Es(S), \text{MASK}), \quad \mathcal{L}_{res} = \sum_{j=1}^{V}\sum_{i=1}^{T} -\log p_\phi(t_i^j|t_i^{1:j-1}, Es(S), j). \quad (4)$$

In this formulation, $\mathcal{L}_{mask}$ predicts the masked motion tokens $t_i$ at each time step $i$ based on the input audio and the special [MASK] token. Conversely, $\mathcal{L}_{res}$ focuses on learning from multiple quantization layers, where $t_i^j$ represents the motion token from quantizer layer $j$ and $t_i^{1:j-1}$ includes the tokens from preceding layers. We also feed the predicted tokens into the RVQ decoder for gesture reconstructions, with velocity and acceleration losses Tevet et al. (2022); Siyao et al. (2022).

### 4.3 STRUCTURE-AWARE IMAGE REFINEMENT

To transfer gesture generation to pixel-level video synthesis, we leverage TPS Zhao & Zhang (2022) to achieve portrait animation based on gesture pattern keypoints from Sec. 4.2 through image warping. To address the uncertainties by optical-flow-based deformation, particularly in large motion regions such as the hands and shoulders, we propose a Semantic-Aware Generator. Auto-Link He et al. (2023) demonstrates that the learning of keypoint connections for image reconstruction aids the model in understanding image semantics. Based on this, we leverage keypoint connections as semantic guidance for image refinement.

**Learnable Edge Heatmaps.** Using the gesture motion keypoints, we establish linkages between them to provide structural information. To optimize computational efficiency, we limit the number of keypoint connections to those defined by body joint relationships Wan et al. (2017), rather than considering all potential connections in He et al. (2023).

For two keypoints $\boldsymbol{k}_i$ and $\boldsymbol{k}_j$ within predefined connection groups, we create a differentiable edge map $\boldsymbol{S}_{ij}$. This edge is modeled as a Gaussian function extending along the line connecting the keypoints. Formally, the edge map $\boldsymbol{S}_{ij}$ for keypoints $(\boldsymbol{k}_i, \boldsymbol{k}_j)$ is defined as:

$$\boldsymbol{S}_{ij}(\boldsymbol{p}) = \exp\left(v_{ij}(\boldsymbol{p})d_{ij}^2(\boldsymbol{p})/\sigma^2\right), \quad (5)$$

where $\sigma$ is a learnable parameter controlling the edge thickness, and $d_{ij}(\boldsymbol{p})$ is the $L_2$ distance between the pixel $\boldsymbol{p}$ and the edge defined by keypoints $\boldsymbol{k}_i$ and $\boldsymbol{k}_j$:

$$\boldsymbol{d}_{ij}(\boldsymbol{p}) = \begin{cases} \|\boldsymbol{p} - \boldsymbol{k}_i\|_2 & \text{if } t \leq 0, \\ \|\boldsymbol{p} - ((1-t)\boldsymbol{k}_i + t\boldsymbol{k}_j)\|_2 & \text{if } 0 < t < 1, \quad \text{where} \quad t = \frac{(\boldsymbol{p} - \boldsymbol{k}_i) \cdot (\boldsymbol{k}_j - \boldsymbol{k}_i)}{\|\boldsymbol{k}_i - \boldsymbol{k}_j\|_2^2}. \quad (6) \\ \|\boldsymbol{p} - \boldsymbol{k}_j\|_2 & \text{if } t \geq 1, \end{cases}$$

Here, $t$ denotes the normalized distance between $\boldsymbol{k}_i$ and the projection of $\boldsymbol{p}$ onto the edge.

To derive the edge map $\boldsymbol{S} \in \mathbb{R}^{H \times W}$, we take the maximum value at each pixel across all heatmaps:

$$\boldsymbol{S}(\boldsymbol{p}) = \max_{ij} \boldsymbol{S}_{ij}(\boldsymbol{p}). \quad (7)$$

We generate heatmaps at various resolutions. Inspired by SPADE Park et al. (2019), we treat these structural heatmaps as semantic guidance for image generation. A U-Net with residual blocks utilizes spatial semantic control from the edge heatmaps to refine the final video output.

**Training Objective.** We employ a conditional adversarial loss Mirza & Osindero (2014), along with perceptual similarity loss Johnson et al. (2016) and $L1$ loss for image refinement. The

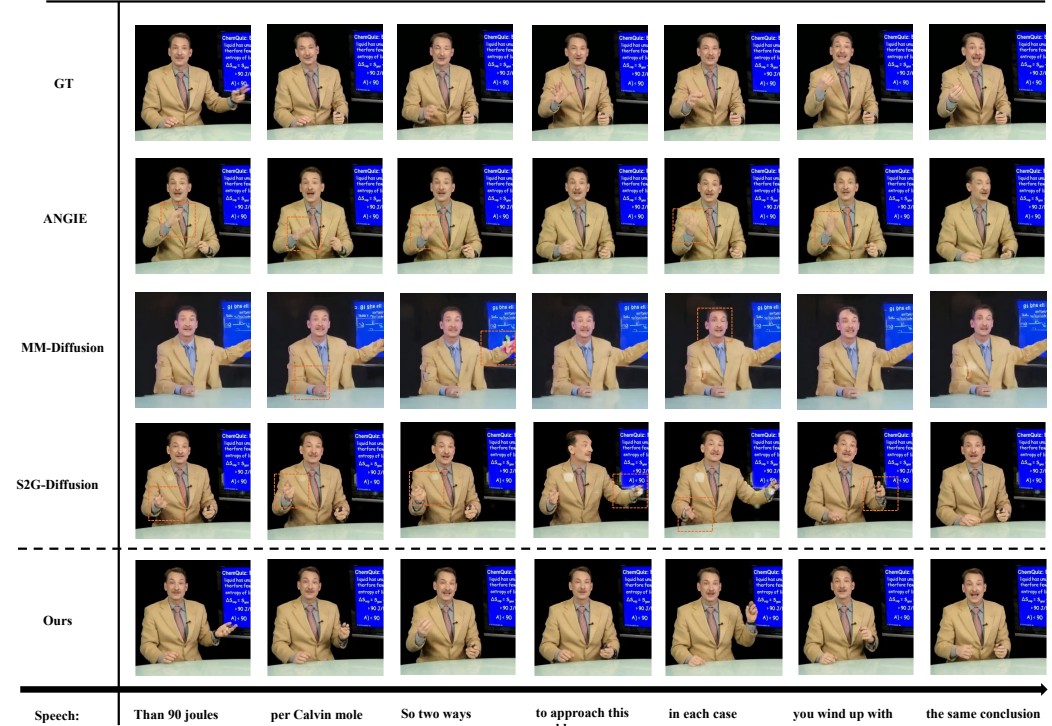

Figure 4: **Visual comparisons.** Our method generates high-quality hand and shoulder motions, and presents metaphoric gestures when saying *"90 joules,"* and *"in each case."*

discriminator utilizes the edge heatmap as a condition to compare generation against ground truth:

$$\mathcal{L}_{cGAN}(G, D) = \mathbb{E}_{I_{gt}, map} \left[ \log D\left(I_{gt}, map\right) \right] + \mathbb{E}_{map} \left[ \log\left(1 - D\left(map, I_{gan}\right)\right) \right], \quad (8)$$

where $map$ denotes the edge heatmap, and $(\cdot, \cdot)$ indicates concatenation.

## 5 EXPERIMENTS

Since our work focuses on joint gesture motion and video generation, the main experiments primarily compare our approach with existing methods that also address joint generation. Comparisons specifically for gesture motion generation and avatar video rendering are deferred to the Appendix, where they are treated as separate, disentangled modules and compared with relevant works.

### 5.1 EXPERIMENTAL SETTINGS

**Dataset and preprocessing.** We utilize PATS Ginosar et al. (2019); Ahuja et al. (2020) for the experiments. It contains 84,000 clips from 25 speakers with a mean length of 10.7s, 251 hours in total. For a fair comparison, following the literature Liu et al. (2022c); He et al. (2024) and replace the missing subject, with 4 speakers are selected (*Noah*, *Kubinec*, *Oliver*, and *Seth*). All video clips are cropped with square bounding boxes, centering speaks, resized to $256 \times 256$. We defer the additional details in the Appendix. After filtering, we obtain around 1000 clips for each speaker, randomly divided into 90% for training and 10% for evaluation, 4,000 in total.

**Baseline Methods.** We benchmark Realistic-Gesture against several co-speech gesture video generation methods: (1) ANGIE Liu et al. (2022c), a work in co-speech gesture video synthesis; (2) MM-Diffusion Ruan et al. (2023), an audio-video generation model demonstrated on the AIST++ dataset Li et al. (2021) that produces audio-driven human motion videos; and (3) S2G-Diffusion He et al. (2024), the most recent advancement in this domain. Notably, due to MM-Diffusion's fixed generation of 34 frames, we segment the audio accordingly for each generation.

### 5.2 QUANTITATIVE EVALUATION

**Evaluation Metrics.** We evaluate gesture motion and pixel-level video quality separately. For gesture motion metrics, we use **Fréchet Gesture Distance (FGD)** Yoon et al. (2020) to measure

Table 1: Quantitative results on the test set. Bold indicates the best performance. Our method performs better in terms of both gesture motions and video generation quality.

| Name | Gesture-Motion Evaluation | | | | Video Quality Assessment | | |
|---|---|---|---|---|---|---|---|
| | FGD $\downarrow$ | Div. $\uparrow$ | BAS $\uparrow$ | PCM $\uparrow$ | FVD $\downarrow$ | $VQA_A \uparrow$ | $VQA_T \uparrow$ |
| Ground Truth (GT) | 0.0 | 14.012 | 1.00 | 1.000 | 0.0 | 95.694 | 5.329 |
| ANGIE | 67.524 | 6.674 | 0.783 | 0.372 | 526.247 | 88.144 | 4.729 |
| MM-Diffusion | 137.62 | 3.212 | 0.646 | 0.112 | - | 79.561 | 4.238 |
| S2G-Diffusion | 23.646 | 10.848 | 0.974 | 0.447 | 486.134 | 93.553 | 5.401 |
| Ours | 1.303 | 13.260 | 0.996 | 0.572 | 476.120 | 96.326 | 6.081 |

the distribution gap between real and generated gestures in feature space, **Diversity (Div.)** Lee et al. (2019) to calculate the average feature distance between generated gestures, **Beat Alignment Score (BAS)** following Li et al. (2021), and **Percent of Correct Motion parameters (PCM)**, difference of generation deviate from ground-truth following Chen et al. (2024). We extract 2D human poses for face and body using MMPose OpenMMLab (2020), which differs from S2G-Diffusion that focuses solely on body poses. The first two metrics utilize an auto-encoder trained on PATS poses.

For pixel-level video quality, we assess **Fréchet Video Distance (FVD)** Unterthiner et al. (2018) for the overall quality of gesture videos, $VQA_A$ for aesthetics and $VQA_T$ for technical quality based on Dover Wu et al. (2023), pretrained on a large-scale dataset with labels ranked by real users.

**Evaluation Results.** We present quantitative evaluations in Tab. 1. Our approach significantly outperforms existing methods in gesture motion metrics, achieving an FGD of **1.303** and a Diversity score of **13.260**. These results indicate that our generated gesture patterns closely resemble those from ground-truth videos, exhibiting both high naturalness and a broader range of motion patterns. For video quality assessment, we use the FVD metric to evaluate the similarity of the generated video distribution to the ground-truth videos. Our model achieves the lowest FVD among the compared methods, demonstrating superior performance. The $VQA_A$ and $VQA_T$ metrics measure perceived user preferences for video generation content. Notably, our approach yields a $VQA_A$ of **96.326** and a $VQA_T$ of **6.081**, surpassing the ground-truth videos. This success can be attributed to our structure-aware image enhancement design. In contrast, MM-Diffusion produces limited gesture patterns due to its design, which generates only a few continuous frames and struggles to learn diverse motion patterns from speech audio. ANGIE leverages MRAA Siarohin et al. (2021) for regional coarse motion patterns but lacks the precision necessary for motion control aligned with speech, resulting in low diversity and beat alignment. S2G-Diffusion performs better than ANGIE but still fails to generate fine-grained gesture patterns, as it relies on image optical flows without adequately focusing on the nuances of human facial and body movements.

### 5.3 QUALITATIVE EVALUATION

**Evaluation Results.** We provide qualitative evaluations of video generation in Fig. 4. MM-Diffusion generates unrealistic shoulder movements. ANGIE produces misaligned gesture motions with the accompanying speech. Although S2G-Diffusion shows improvement over ANGIE, it struggles with local

Table 2: Subjective evaluation results are shown as Mean Opinion Scores (MOS).

| Methods | $MOS_1$ | $MOS_2$ | $MOS_3$ | $MOS_4$ |
|---|---|---|---|---|
| | User Study | | | |
| GT | 4.7 | 4.7 | 4.7 | 4.65 |
| MM-Diffusion | 1.35 | 1.65 | 1.4 | 1.55 |
| ANGIE | 1.95 | 3.25 | 1.9 | 2.25 |
| S2G-Diffusion | 3.0 | **3.6** | 3.15 | 3.0 |
| Ours | **3.35** | 3.05 | **3.35** | **3.25** |

regions, such as the hands, due to its reliance on unsupervised keypoints for global transformations, which neglects local deformations. In contrast, our method demonstrates high-quality video generation, particularly in the facial and body areas. The alignment between gesture and speech is notably enhanced through our speech-content-aware gesture latent representation. For example, when the actor says *"90 joules,"* he points to the screen, and he emphasizes phrases like *"so two ways"* and *"in each case"* by raising his hands as metaphoric gestures. This coordination exemplifies our approach's capability to produce contextually relevant and expressive gestures.

**User Study.** We conducted a user study to evaluate the visual quality of our method. We sampled 80 videos from each method and ground-truth, and invited 20 participants to conduct Mean Opinion Scores (MOS) evaluations. The rating ranges from 1 (poorest) to 5 (highest). Participants rated the videos on: (1) $MOS_1$: *"How **realistic** does the video appear?"*, (2) $MOS_2$: *"How **diverse** does the gesture pattern present?"*, (3) $MOS_3$: *"Are speech and gesture **synchronized** in this video?"* and (4) $MOS_4$: *"What is your **overall evaluation** of the video"*. The videos were presented in random order

Table 3: Ablations of our method. We exam the keypoint design, gesture representation, gesture generator architecture, training & inference strategy and image-refinement. Bold indicates the best performance.

| Kp Repr. | FVD↓ | LPIPS↓ | PSNR↑ |
|---|---|---|---|
| Unsup-kp | 387.05 | 0.05 | 27.41 |
| 2D-pose | 272.18 | 0.05 | 27.26 |
| + flex kp | 377.14 | 0.06 | 25.36 |
| full-model | 225.77 | 0.04 | 27.17 |

(a) Configurations for keypoint design.

| G-Repr. | FGD↓ | Div.↑ | PCM↑ |
|---|---|---|---|
| baseline | 262.675 | 18.142 | 0.279 |
| +RVQ | 34.940 | 6.713 | 0.327 |
| + separate | 21.473 | 10.536 | 0.412 |
| + distill | 1.303 | 13.260 | 0.582 |

(b) Gesture motion representations.

| G-Gen. | FGD↓ | Div.↑ | PCM↑ |
|---|---|---|---|
| w/o res | 3.372 | 11.359 | 0.513 |
| concat | 3.415 | 11.314 | 0.514 |
| w/o align | 8.382 | 11.452 | 0.373 |
| full-model | 1.303 | 13.260 | 0.582 |

(c) Generator archiecture design.

| Refine | VQA$_A$↑ | VQA$_T$↑ | FVD↓ |
|---|---|---|---|
| w/o refine | 91.248 | 5.381 | 492.341 |
| + UNet | 93.958 | 5.479 | 484.323 |
| + skeleton | 95.902 | 5.479 | 475.636 |
| + heatmap | 96.326 | 6.081 | 476.120 |

(d) Image-refinement strategies.

| M-Ratio | FGD↓ | Div.↑ | PCM↑ |
|---|---|---|---|
| Uni 0-1 | 3.348 | 14.312 | 0.513 |
| Uni .3-1 | 3.232 | 12.58 | 0.512 |
| Uni .5-1 | 1.303 | 13.260 | 0.582 |
| Uni .7-1 | 1.790 | 13.49 | 0.572 |

(e) mask-ratio during training.

| iter. | FGD↓ | Div.↑ | PCM↑ |
|---|---|---|---|
| 5 | 1.303 | 13.260 | 0.582 |
| 10 | 1.642 | 13.40 | 0.575 |
| 15 | 1.828 | 13.49 | 0.573 |
| 20 | 1.881 | 13.49 | 0.572 |

(f) Mask decoding steps.

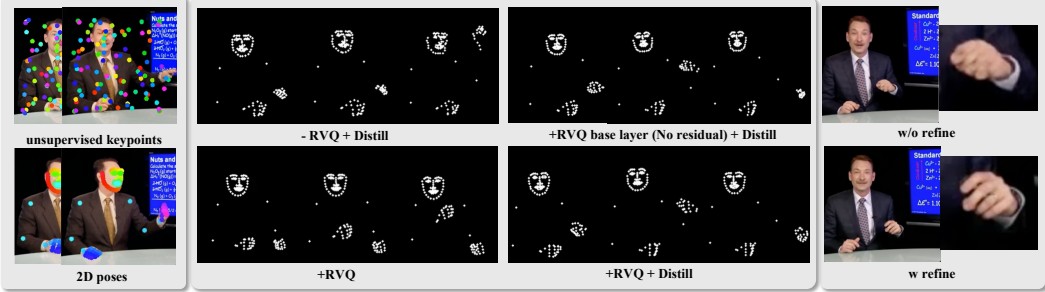

Figure 5: **Ablation visualizations.** Left: motion by unsupervised keypoints or 2d poses; Middle: RVQ-based gesture representation and generation; Right: image-refinement helps hand generation.

to capture participants' initial impressions. As shown in Tab. 2, our method outperformed others across realness, synchronization, and overall quality, but lower in diversity than S2G-Diffusion. We defer additional details of the user study in the appendix.

## 5.4 ABLATION STUDY

In this section, we present ablation study of keypoint design for image warping, gesture pattern representation exploraton, gesture generator architecture design, and varios comparisons of image-refinement. We defer additional experiments in the Appendix.

**Motion Keypoint Design** We evaluate three keypoint representations for image-warping: (1) unsupervised keypoints for global optical-flow transformation (as in ANGIE and S2G-Diffusion), (2) 2D human poses, and (3) 2D human poses augmented with flexible learnable points. Each design is assessed using TPS Zhao & Zhang (2022) transformation, with self-reconstruction based on these keypoints for evaluation. We compare the full-model reconstruction with refinement against the first three designs without refinement. As shown in Tab. 3a, learnable keypoints lead to a significant decrease in FVD, highlighting their inadequacy for motion control. The 2D landmark-based keypoints yield slightly lower SSIM scores, likely due to their limited capacity to represent global transformations. The inclusion of flexible keypoints does not enhance the image-warping outcomes. Consequently, we opt to utilize 2D pose landmarks exclusively for our study.

**Motion Representation.** We evaluate several configurations: (1) baseline: no motion representation, relying solely on the generator to synthesize raw 2D landmarks; (2) + RVQ: utilizing Residual VQ (RVQ) to encode joint face-body keypoints; (3) + separate motion: employing two RVQs for independent face and body motions; (4) + distill: learning joint embeddings for speech and gesture in both face and body motions, followed by distillation for RVQ tokenization. We discover RVQ significantly improve the precise pose location while distillation leads to natural movements.

Figure 6: Our model supports multiple video gesture generation end editing applications.

**Generator Design.** We explore various designs for the gesture generator: (1) w/o res: no residual gesture decoder; (2) concat: instead of using cross-attention for audio control, we concatenate the audio features with gesture latent features element-wise during generation; (3) w/o align: the audio encoder is randomly initialized rather than initialized from face and body contrastive learning. Our findings indicate that the Residual Gesture Generator significantly enhances finger motion generation. The cross-attention design outperforms element-wise concatenation, while the pre-alignment of the audio encoder notably improves FGD. We attribute this improvement to the shared similarities in the codebook and audio encoders during contrastive alignment.

**Image Refinement.** We examine various network designs for motion generation, specifically: (1) w/o refine: no image refinement, relying solely on image warping; (2) + UNet: employing a standard UNet; (3) + pose skeleton: integrating connected skeleton maps as in the diffusion ReferenceNet Hu et al. (2023); (4) + edge heatmap: substituting the previous design with our learnable edge heatmap. Our experiments reveal that the edge heatmap outperforms skeleton maps, likely due to the learnable thickness of connections, which provides better semantic-aware generation guidance.

**Training and Inference Strategy.** We evaluate the mask ratio during training and the number of inference steps during decoding. As shown in Tab. 3f, our model requires only 5 inference steps, in contrast to over 50 or 100 steps in diffusion-based models. Furthermore, a uniform masking ratio between 0.5 and 1 during training yields optimal performance.

### 5.5 APPLICATION

**Long Sequence Generation.** Shown on the left of Fig. 6, to generate long sequences, we start with the initial frame and the corresponding target audio, which we segment into smaller windows. After generating the first segment, we use the last few frames of the generated output as the new starting frame conditions for the next segment of audio, allowing for a iterative outpainting.

**Video Gesture Editing.** For gesture editing and inpainting, we first extract the keypoints from a given video sequence and tokenize the face and body movements into motion tokens. Thanks to the model's bidirectional decoding capability, we insert [MASK] tokens wherever edits are needed. Trained on temporal masking, the model generate coherent gestures in the masked areas. By incorporating different speech input and speaker embeddings, we can create new gesture patterns and re-render the video based on the edited latent motion tokens.

**Gesture Pattern Transfer.** Given the design of our framework, with different identity embedding, the model can generate different gesture patterns given the input identity embedding control given the same audio. Please see the demo videos in our Appendix for more details.

### 6 CONCLUSION

We present **Realistic-Gesture**, a framework for generating realistic co-speech gesture videos. To ensure the gestures cohere well with speech, we propose speech-content aware gesture motion representation though knowledge distillation from the gesture-speech aligned features obtained through contrastive learning. Our masked gesture motion generator enables the creation and editing of long, high-quality gesture motion sequences. Our pixel-level refinement module further improves the transformation of inferred gesture motions into realistic animations for large-scale body motion. We believe this work will encourage further exploration of the relationship between gesture patterns and speech context for more compelling gesture video generations in the future.

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
