# OpenReview forum: "Realistic-Gesture: Co-Speech Gesture Video Generation through Semantic-aware Gesture Representation"
_ICLR.cc/2025/Conference — Submitted to ICLR 2025_

### Official Review · Reviewer_XVKk · 2024-10-27

**Soundness:** 3
**Presentation:** 3
**Contribution:** 3
**Rating:** 8
**Confidence:** 5

**Summary:**

This paper improves two-stage co-speech video gesture generation method in two main aspects. Two-stage here is auido2pose and pose2video stages. The authors:

**1. Improve audio2pose stage by contrastive pretraining.**

- The baseline started from a mask represenation learning and Residual VQ-VAE, refer to MoMask [Guo et al. 2024]. The authors modify it to fuse audio feature by cross-attention.
- The authors pretrain gesture encoders, and audio encoders using a audio2pose contrastive learning. Then, the audio encoder in audio2pose generation stage is initialized by pretrained audio encoder. The gesture encoder, in generation stage is distilled to keep the similarity to pretrained gesture encoder.

**2. improve pose2video stage by learned edge-map for image warping.**

- The baseline is a pixel-level image-warping based pipeline. Using Thin Plate Splines (TPS) as mentioned in Appendix. G. The image was initially warpped and then refined by network.
- The authors propose a thickness learnable edge heatmap. And using this heatmap to improve the warping results.

The authors present experiments to show the overall results outperform previous methods, and have the ablation studies for demonstrating each improvement is effective.

**Strengths:**

The proposals in this paper are correct and have insights:

**1. Improve audio2pose stage by contrastive pretraining.**

The concept of improving the audio2pose generation via contrastive learning pretraining is correct and from the results, it improves the audio2pose generation significantly.  In image generation field, researchers use pretrained text-image CLIP encoder to improve the results. this paper shares the same insight in audio2pose domain.

**2. improve pose2video stage by learned edge-map for image warping.**

This is a valuable improvement to pixcel-level image warping based approach. Firstly, most of current methods may focus on the improvement of latent-diffusion based methods, but they are typically slow in real-world applications, and have noisy background which require further post-processing. The pixcel-level approach, is very valuable for improvement due to the clean background and faster inference speed. Second, it is correct the most important problem is the correctness of "flow" for warpping based approach, and the thickness of flow is the most important factor. using a learned approach to solve this avoid a lot of manually parameter adjustments.

The experiments, in particular for the ablation part, show the effectiveness of each proposed module.

**Weaknesses:**

I have a few unclear implementation details.

1. Random mask for contrastive learning training. in Line 211-212. "we random mask 30% ..." I'm confused the mask is in a continues short-sentence level or frame-level. and why it could improve the low-level similarity learning. I suggest writing more explainations here.

**Questions:**

The questions will influence my score is:

1. Random mask for contrastive learning training. in Line 211-212. "we random mask 30% ..." I'm confused the mask is in a continues short-sentence level or frame-level. and why it could improve the low-level similarity learning. I suggest writing more explainations here.

2. What is the sequence length used for training the contrastive learning and audio2pose generation?

3. Similarly, what is the resolution for the image training and evaluation stage, will the image-warping based approach suffer a GPU memory issue when resolution increase?

The question will not influence my score is:

1. some discussion about early stage audio2pose + pose2video work like speech2gestures [Ginoar et al. 2019] and Speech Driven Template [Qian et al. 2021]. in related work section.

---

> ### Author Response · Authors · 2024-11-14
>
> Thanks for your valuable suggestions and comments to this work. We address some of the technical details as follows:
>
> ### Reply to Masking Strategy and Clarification:
> Thank you for pointing this out. To clarify, for an 80-frame video, we randomly select 24 continuous frames to mask during training. We conducted a preliminary experiment to assess the effect of random masking by comparing it with a setting that did not use any masking. The results are summarized in the table below. We include this "no mask" setting in the manuscript, in addition to the two original settings described in the Appendix.
>
> ### Speech-Face retrieval vs. Face-Speech retrieval
> | **Setting**          | **R@1 ↑**  | **R@2 ↑**  | **R@3 ↑**  | **R@5 ↑**  | **R@10 ↑** | **R@1 ↑**  | **R@2 ↑**  | **R@3 ↑**  | **R@5 ↑**  | **R@10 ↑** |
> |----------------------|------------|------------|------------|------------|------------|------------|------------|------------|------------|------------|
> | **(a) All**          | 0.181      | 0.350      | 0.485      | 0.722      | 1.343      | 0.226      | 0.361      | 0.429      | 0.677      | 1.207      |
> | **(a) w/o mask**     | 0.142      | 0.326      | 0.388      | 0.656      | 1.112      | 0.158      | 0.299      | 0.343      | 0.612      | 1.026      |
> | **(b) Small batches**| 26.230     | 45.318     | 59.330     | 77.019     | 89.858     | 24.977     | 44.822     | 59.894     | 77.775     | 90.264     |
> | **(b) w/o mask**     | 25.373     | 44.221     | 60.432     | 78.141     | 88.232     | 24.534     | 44.532     | 59.121     | 74.232     | 87.675     |
>
> ### Speech-Body retrieval vs. Body-Speech retrieval
> | **Setting**          | **R@1 ↑**  | **R@2 ↑**  | **R@3 ↑**  | **R@5 ↑**  | **R@10 ↑** | **R@1 ↑**  | **R@2 ↑**  | **R@3 ↑**  | **R@5 ↑**  | **R@10 ↑** |
> |----------------------|------------|------------|------------|------------|------------|------------|------------|------------|------------|------------|
> | **(a) All**          | 0.102      | 0.237      | 0.327      | 0.587      | 1.230      | 0.158      | 0.271      | 0.406      | 0.654      | 1.320      |
> | **(a) w/o mask**     | 0.112      | 0.143      | 0.303      | 0.494      | 1.023      | 0.144      | 0.253      | 0.384      | 0.599      | 1.187      |
> | **(b) Small batches**| 25.542     | 43.660     | 57.954     | 77.471     | 90.309     | 24.052     | 43.874     | 58.495     | 76.986     | 89.745     |
> | **(b) w/o mask**     | 23.437     | 40.653     | 54.332     | 74.983     | 88.273     | 22.454     | 40.235     | 56.383     | 74.436     | 88.675     |
>
>
> From these results, we conclude that global contrastive learning can be enhanced by incorporating simple random masking, which helps improve the temporal alignment between audio and gesture features.
>
> ---
>
> ### Reply to Technical Details (Sequence Length and Audio2Pose Generation):
> For both contrastive learning and the audio2pose generation tasks, we use 80-frame videos, which correspond to approximately 3.2 seconds of video. This information is also availble in Appendix, Implementation-Detail Section.
>
> ---
>
> ### Reply to Model Efficiency (Resolution and GPU Memory):
> Thank you for raising this point. We’ve added additional details in the table below, along with a comparison to the resource consumption of other models. For reference, we provide a link to our anonymous webpage demonstrating the comparison: [here](https://anonymousaisubmission.github.io/). The visual comparisons with Animate Anyone will be uploaded by next two days.
>
> ### Resource Consumption Comparison with Stable-Diffusion-based Image-Animation Models (1 NVIDIA A100 GPU), Inference for 80-frame Videos
>
> * denotes our re-implementation on the PATS dataset.
>
> | **Methods**       | **Training ↓** | **Batch Size** | **Resolution** | **Memory ↓** | **Training Task**  | **Inference ↑** |
> |-------------------|----------------|----------------|----------------|--------------|-------------------|------------------|
> | AnimateAnyone*    | 10 days        | 4              | 512            | 40 GB        | Pose-2-Img        | -                |
> | AnimateAnyone*    | 5 days         | 2              | 512            | 42 GB        | Img-2-Vid (24 frames)         | 15s              |
> | **Ours**          | 2.5 days       | 64             | 256            | 64 GB        | Img-Warp          | ≤ 1s             |
> | **Ours**          | 1 day          | 64             | 256            | 48 GB        | Img-Refine        | ≤ 1s             |
> | **Ours**          | 3.5 days       | 32             | 512            | 60 GB        | Img-Warp          | ≤ 1s             |
> | **Ours**          | 1 day          | 32             | 512            | 40 GB        | Img-Refine        | ≤ 1s             |
>
> ---

---

> ### Comment · Reviewer_XVKk · 2024-11-17
> **Thank you for the feedback**
>
> Thank you for the detailed feedback, the experiments with object scores addressed my concerns about the usage of random mask, training length and image resolution,
>
> from the additional experiments, I think it is valuable that showing mask representation learning could further improve the performance of contrastive learning. That may inspire further work to combine these two.
>
> And the inference speed comparison demonstrates that this non-diffusion-based model is more practice for real-world applications such as real-time generation or streaming systems.  Even though a lot of work is based on diffusion, I think the exploration of the non-diffusion models still be valuable for these online applications.  so I raised my score from 6 to 8.

---

### Official Review · Reviewer_zy2t · 2024-11-02

**Soundness:** 3
**Presentation:** 3
**Contribution:** 3
**Rating:** 6
**Confidence:** 4

**Summary:**

The paper introduces a framework aimed at generating realistic co-speech gesture videos. This framework tackles the challenges of establishing correspondences between speech signals and body movements, inferring suitable gestures from speech samples, and rendering the target speaker performing these gestures in a lifelike manner. The authors propose three innovative components to accomplish this:
1. A speech-aware gesture representation that aligns facial and body gestures with speech semantics, enabling fine-grained control.
2. A mask gesture generator that maps audio signals to gestures by predicting masked motion tokens, thereby enabling bidirectional and contextually relevant gesture synthesis and editing.
3. A structure-aware refinement module that uses a multilevel, differentiable edge connection to link gesture keypoints for the generation of detailed videos.

The paper's contributions include the production of highly realistic, speech-aligned gesture videos, as well as support for long-sequence generation and video gesture editing applications. The experiments demonstrate the method's superiority over existing approaches in both quantitative and qualitative metrics.

**Strengths:**

The paper presents a well-articulated approach to generating co-speech gesture videos, demonstrating a clear understanding of the associated challenges and the necessity for a sophisticated method to address them.
- The authors have crafted a compelling narrative surrounding the design of their method, offering adequate motivation for each component.
- From a visual standpoint, the proposed method surpasses existing state-of-the-art techniques.
- The integration of speech-aware gesture representation, masked gesture generation, and structure-aware refinement is a significant strength, enhancing both the realism and controllability of gesture synthesis.

**Weaknesses:**

**Long Sequence Generation Quality**: The author identifies the capacity to generate long sequences as a primary contribution of their method. However, the supplementary video material reveals that the quality of long sequence generation is subpar; in later stages, the hands become blurry, and behavioral patterns tend to repeat. Furthermore, the author provides a limited number of videos and fails to present quantifiable metrics for assessing the quality of long sequence generation. Consequently, I have significant doubts regarding the framework's ability to generate long sequences effectively.

**Lack of Novelty in Core Components**: Although the framework offers an integrated approach to gesture synthesis, its core components—speech-aware gesture representation and masked gesture generation—have been examined in various forms in recent literature.
- The contrastive learning approach for speech-gesture alignment has been utilized in gesture synthesis methods [1,2,3] and is very similar to the CSMP introduced in [1].
- Similarly, the use of RVQ [4,5] and masked gesture synthesis techniques [6] is not new to the field.

The paper should offer a more detailed comparison with these methods and present a more comprehensive review of the related works.

[1] Deichler, Anna, et al. "Diffusion-based co-speech gesture generation using joint text and audio representation." Proceedings of the 25th International Conference on Multimodal Interaction. 2023.

[2] Liu, Xian, et al. "Learning hierarchical cross-modal association for co-speech gesture generation." Proceedings of the IEEE/CVF Conference on Computer Vision and Pattern Recognition. 2022.

[3] Xu, Zunnan, et al. "Chain of generation: Multi-modal gesture synthesis via cascaded conditional control." Proceedings of the AAAI Conference on Artificial Intelligence. Vol. 38. No. 6. 2024.

[4] Zhang, Zeyi, et al. "Semantic Gesticulator: Semantics-Aware Co-Speech Gesture Synthesis." ACM Transactions on Graphics (TOG) 43.4 (2024): 1-17.

[5] Wang, Congyi. "T2M-HiFiGPT: Generating High Quality Human Motion from Textual Descriptions with Residual Discrete Representations." arXiv preprint arXiv:2312.10628 (2023).

[6] Mao, Xiaofeng, et al. "MDT-A2G: Exploring Masked Diffusion Transformers for Co-Speech Gesture Generation." Proceedings of the 32nd ACM International Conference on Multimedia. 2024.

**Questions:**

- This method utilizes 2D keypoints as guiding conditions. However, variations in human skeletal structure and facial shape may result in distortions of the generated portrait identity. Has alignment been performed in this context? If so, what techniques are employed for alignment?
- What is the quality of generation for long sequences? Can the author provide quantitative indicators to support their findings?
- Can the author provide a more detailed description of the differences compared to the literature referenced in the Weakness? If the author can offer sufficient explanations to enhance the novelty of the paper, I will consider increasing the score.

---

> ### Author Response · Authors · 2024-11-14
>
> ### Response to Concerns About Novelty in Core Components:
>
> We appreciate the reviewer’s feedback and recognize that the **VQ + Masking-based Representation Learning** for gesture generation may seem similar to existing work. However, the primary novelty of our method lies in **contextual distillation through contrastive alignment** for learning speech triggers that drive gesture patterns, as discussed in Section 4.1. To the best of our knowledge, no prior work has explored the **benefits of audio-motion representation learning for constructing contextualized motion representations to achieve better gesture generation**.
>
> To further highlight the contribution of our approach, we plan to add additional experiments [here](https://anonymousaisubmission.github.io/) comparing our method with other approaches focused solely on gesture pose generation (without video synthesis), including:
>
> 1. **EMAGE**: Towards Unified Holistic Co-Speech Gesture Generation via Expressive Masked Audio Gesture Modeling
> 2. **TalkSHOW**: Generating Holistic 3D Human Motion from Speech
> 3. **Rhythmic Gesticulator**: Rhythm-Aware Co-Speech Gesture Synthesis with Hierarchical Neural Embeddings (as suggested by Reviewer P3xu)
>
> These works utilize VQ+masking-based models or contrastive domain alignment, making them relevant baselines for comparison. We will conduct these experiments using the **BEAT-X dataset** for a fair comparison. For consistency, we will exclude the image-animation component from our method and extend gesture representation from 2D to 3D poses, incorporating SMPL-X expressions for facial gestures, as done in the literature.
>
> ---
>
> ### Response to Pose Alignment:
>
> We apologize for any confusion caused by our previous explanation. To address variations in human skeletal structure, the pose representation for a sequence is calculated as the difference in position (Δx, Δy) relative to the coordinates of the starting frame, combined with the original position of that frame. This is represented as [Δx, Δy, x₁, y₁]. This formulation ensures that pose alignment is achieved since all subsequent frames’ motions are conditioned on the starting frame’s pose.
>
> ---
>
> ### Response to Detailed Comparison:
>
> Regarding speech-gesture alignment:
>
> In works like [1] and [2], contrastive learning is employed to align audio and gesture modalities, initializing the audio encoder for gesture generation. However, our method differs in the following key ways:
>
> 1. **Contextualized Motion Representation**: We directly encode alignment information into the motion representation using an RQ-codebook. This allows the semantics and contextual triggers from speech (e.g., pronouns like “this” or “they”) to be fused into the motion embedding. This enables the generator to easily identify the corresponding motion representation in response to speech triggers during generation. In contrast, [1] and [2] lack such a strategy for creating more contextualized motion representations.
>
>    Our ablation study in Table 3 (b), (c) shows that, with modality alignment alone, we achieve an FGD score of around 21. However, applying contextual distillation through contrastive learning further refines the motion representation, leading to better contextual-aware generation.
>
> 2. **Temporal Alignment**: In addition to high-level, global sequence alignment, we also address **temporal alignment** by applying temporal masking and audio beat classification. None of the other works consider this intricate temporal alignment between speech and gestures.
>
> For **VQ + Masking-based Generation**:
> Works such as [4], [5], and [6] explore VQ-quantization with autoregressive, or diffusion-based generation techniques. While these methods do utilize VQ-based frameworks, we believe our contribution in the generator design is unique in that we apply a **Iterative re-masking-based approach specifically to co-speech gesture generation**.
>
> Generating gestures as long sequences presents a challenge for autoregressive or diffusion models, as they are often too slow for real-time generation. In contrast, our approach using masking-based generation can efficiently generate gestures in just **5 steps**, with further steps potentially degrading quality by disrupting the temporal relationships between the modalities (different from text-motion domain where more step masking will be more beneficial in [5). Our extensive experiments in Tab. (e) and (f) in generator design and generation procedures is non-trival but presenting results potentially be helpful for future.
>
> Additionally, while [3] shares the same domain (co-speech gesture generation), it does not present overlapping or similar novelties to our work.

---

> > ### Author Response · Authors · 2024-11-18
> >
> > ### Reply to Long Sequence Generation Quality:
> > Thank you for pointing this out. We will provide additional statistics to measure the performance degradation during long sequence generation and editingin the revised supplementary material or can be found [here](https://anonymousaisubmission.github.io/). During our additional study of gesture generation on BEAT-X dataset, we found out long sequence generation, unlike for PATS dataset, does not affect the generation quality much. We suppose this might be due to PATS dataset are sources with mostly less than 10-second videos.
> >
> > ---
> >
> > ### Reply to Detailed Comparisons:
> > We in addition provide the experiments for BEAT-X dataset in the revised supplementary material with visual comparisons in demo page videos. We discover adding contextual distillation can improve the generation quality and achieve better quality than any existing works. Please see the revised supplementary material for further details.
> >
> > ---
> >
> > We hope our explanations and additional demo videos can provide clear presentations of our work and address your concern.

---

> > > ### Comment · Reviewer_zy2t · 2024-11-22
> > >
> > > I have raised my score from 5 to 6, as the author has addressed most of my concerns. Can the author further provide quantitative indicators to evaluate the method's performance in generating long sequences compared to EMAGE, Talkshow and Rhythmic Gesticulator? The quality difference between the methods presented in the one-minute supplementary video is not significant.

---

> > > > ### Author Response · Authors · 2024-11-23
> > > >
> > > > Thanks for your additional comments. For the table we provided in the revised supplementary material, our measure is conducted on long-sequence generation.
> > > >
> > > >
> > > > | Method                         | FGD (↓) | BC (↑)  | Diversity (↑) | MSE (↓)    | LVD (↓)   |
> > > > |--------------------------------|---------|---------|---------------|------------|-----------|
> > > > | Rhythmic Gesticulator [AO et al. 2022] | 6.453   | 6.558   | 9.132         | -          | -         |
> > > > | TalkSHOW [Yi et al. 2022]       | 6.209   | 6.947   | 13.47         | 7.791      | 7.771     |
> > > > | EMAGE [Liu et al. 2023]         | 5.512   | **7.724**| 13.06         | 7.680      | 7.556     |
> > > > | Ours (w/o Distillation)         | 7.479   | 7.395   | 12.12         | 7.656      | 7.671     |
> > > > | **Ours**                        | **4.650**| 7.370   | **13.55**     | **7.343**  | **7.432** |
> > > >
> > > > To prevent the confusion regarding the evaluation setting, for the short sequence generation setting on BEAT-X. We cut the original testing dataset audios into each segment of 256 frames (about 8.53 seconds) for conducting this evaluation. For the long-sequence generation, we use the raw sequence length of audio from the testing dataset for generation using a sliding window method in the main paper.

---

### Official Review · Reviewer_LNUW · 2024-11-03

**Soundness:** 4
**Presentation:** 3
**Contribution:** 3
**Rating:** 6
**Confidence:** 4

**Summary:**

The paper proposes Realistic-Gesture, a gesture video generation model with speech audio and the first video frame inputs. The proposed method first learns a joint embedding of the speech audio and gesture motions for speech-gesture alignment with CLIP-like contrastive learning. The pose features are 2D face and body landmarks from MMPose. The speech-audio features are the concatenation of WavLM, Mel spectrogram, and beat detections. The gesture motions are tokenized using Residual VQ (RVQ) with the distillation to minimize the cosine similarity with the motion encoder output in the gesture-audio joint embedding. Inspired by VALL-E, the method uses the Masked Gesture Generator for RVQ's base layer and the Residual Gesture Generator for residual layers. The Masked Gesture Generator is a transformer with the cross-attention between the audio embedding and the gesture embedding with AdaIN after the feed-forward layer to condition the model with the speaker identity. The Masked Gesture Generator is trained with randomly masked tokens. The Residual Gesture Generator is similar but consists of embedding layers corresponding to the RVQ residual layers. During training, one residual layer is randomly selected. The inference iteratively predicts the mask probabilities conditioned by the audio embedding to remask the token with the lowest probability for the next iteration. Finally, the source image is warped with TPS [Zhao and Zhang 2022] with the edge maps from the generated keypoints, followed by the image refinement GAN conditioned by the same edge maps, to generate the gesture videos.

**Strengths:**

- Speech-gesture alignment with CLIP-like joint embedding with contrastive learning
- Motion tokenization with RVQ distilled by the joint embedding above
- Audio-conditioned Masked Motion Model with dedicated residual generators for the RVQ residual layers

**Weaknesses:**

The contributions boil down to the motion representation and the motion generation model. There is not much to the image generation module. The visual quality may also be suffering because of the image generation module (see below).

Qualitatively, the generated videos from this method generally look better than other methods in comparison (ANGIE, MM-Diffusion, S2G-Diffusion), especially if I focus on the speech-gesture motions. However, there are moments where the human generation is creepier than others with this method, e.g. in noah1.mp4, the head and the body look disconnected. I am not too surprised that the TPS warping will introduce unnatural human poses, especially with larger 3D motions. Perhaps the image warper and the refiner need more work. The motion representation and the generation model perhaps could be hooked up to some other conditional image/video generator.

The writing is somewhat unclear on the image generation module. The main text directly jumps to the image refinement module without mentioning the TPS image warping, which confused me a bit. Perhaps 4.3 should be titled differently and recap the image warping module while clarifying this is not the paper's contribution (no strong opinion here).

I understand that the space is very limited but I would like more elaborations on the ablations, especially on the motion representation and the generator design, so I can be confident that the main contributions of this paper are indeed effective. See my question.

**Questions:**

Can authors provide videos for the ablation study, especially on the motion representation and the generator designs? I want to be confident that the use of the CLIP-like joint embedding to distill the RVQ for the Masked Motion Model to generate tokens is helping to improve the visual quality.

The speech-gesture joint embedding could perhaps be used to evaluate the alignment of the speech audio and gestures, like the CAPP model from VASA-1 [Xu et al. 2024].

**Details Of Ethics Concerns:**

The dataset seems to be scraped off from YouTube. While the metadata shared in PATS [Ginosar et al. 2019] may be licensed under “CC BY - NC - ND 4.0 International,” I do not think this means the videos linked by the metadata have the same license.

---

> ### Author Response · Authors · 2024-11-18
>
> ### Reply to Speech-Gesture Ablation Videos:
> Thank you for your suggestion. We have provided a webpage with additional ablation studies for your reference: [here](https://anonymousaisubmission.github.io/) or in the revised supplementary material. We additionally included the ablation of distillation as you mentioned for BEAT-X to prevent entanglement of video rendering component affecting your judgement. Additional ablation videos will be updated later as well.
>
> ---
>
> ### Reply to Separate Component Measurement:
> We appreciate your feedback. In response to similar requests from other reviewers, we have decided to separate and more clearly measure the contributions of our work in the following way:
>
> 1. **Gesture Representation and Generation**: We have provided ablation studies based on the BEAT-X dataset to avoid potential unfair comparisons when reimplementing on the PATS dataset.
> 2. **Conditional Video Generation**: In addition, we include the most recent diffusion-based models, **"AnimateAnyone"**, as mentioned in the response to Reviewer P3xu.
>
> We have summarized our experiments and findings in response to Reviewer zy2t.
>
> ---
>
> ### Reply to Speech-Gesture Alignment:
> Thank you for your insightful suggestion. We agree that the Speech-Gesture Alignment model could be used for evaluating the alignment of speech and gestures, similar to the CAPP model from VASA-1. As outlined in Appendix, the model can achieve high recall for speech-gesture retrieval when trained on a set of 32 speech-gesture pairs. Maybe further works can incorporate this component for better co-speech gesture generation evaluation.
>
> ---
>
> ### Reply to Confusion of Abrupt Transition to Image Animation:
> Thank you for pointing this out. We have revised the manuscript to address the abrupt transition to the image refinement module. Specifically, we have clarified the role of the **TPS image warping** and revised Section 4.3 to better explain the image warping process and emphasize that this is not a central contribution of our paper. Please see the updated section in Line.294-297.
>
> ---
>
> We hope our explanations and additional demo videos can provide clear presentations of our work and address your concern.

---

> > ### Comment · Reviewer_LNUW · 2024-11-25
> >
> > The rebuttal videos on ablations are insightful. The alignment makes a more drastic and clear difference in the final result than the distillation. The difference from the distillation is very subtle and whether the difference is an improvement is subjective. Looking at these ablation videos, I am surprised that the FGD with distillation improves significantly in Table 3 (b). Do the authors have an explanation for this?
> >
> > The ablation of BEAT-X and comparison with AnimateAnyone show the advantages of the method.
> >
> > While I am in support of accepting this paper, I do think the paper may want to tone down on the need for distillation, based on the ablation videos. I will keep my score as is.

---

### Official Review · Reviewer_P3xu · 2024-11-03

**Soundness:** 2
**Presentation:** 2
**Contribution:** 3
**Rating:** 5
**Confidence:** 5

**Summary:**

This work proposes Realistic-Gesture, a framework that enhances this process through speech-aware gesture tokenization, a mask gesture generator for audio-to-gesture mapping, and a structure-aware refinement module. The results demonstrate that Realistic-Gesture creates realistic, speech-aligned gesture videos and supports long-sequence generation and editing.

**Strengths:**

This paper proposes a framework for generating co-speech body-gesture videos, dubbed Realistic-Gesture. Realistic-Gesture integrates a speech-aware gesture motion representation, a masked gesture motion generator, and a pixel-level refinement module to facilitate high-quality video generation. The strengths of this work are summarized as follows:
1. The insight of the work is interesting and significantly practical in real-life applications.
2. The experimental workload is solid and persuasive.
3. The demo videos are helpful for reviewers to have a comprehensive understanding of this work.

**Weaknesses:**

However, I still find some weaknesses and some main questions in this manuscript:

1. In the introduction section, the authors put much effort into introducing existing works. This may lead to much overlap with the ``related work'' section. The motivation and insight of this work could be summarized more compactly. Authors can leverage more space to elaborate on the high-level technical contribution of their work.

2. The teaser figure (Fig. 1) seems confused. I cannot obtain significant insight into it. Meanwhile, there are no effective captions in the manuscript.

3. There are some typos, e.g., in line 173, To achieve this goal, We... The letter ``W'' should be lowercase.

4. The technical novelty seems a little bit weak. The Residual Gesture Generator, Masked Gesture Generator are very similar to the previous works, like:

[1] Generating Holistic 3D Human Motion from Speech, in CVPR 2024.

[2] EMAGE: Towards Unified Holistic Co-Speech Gesture Generation via Expressive Masked Audio Gesture Modeling, in CVPR 2024.

[3] BEAT A Large-Scale Semantic and Emotional Multi-Modal Dataset for Conversational Gestures Synthesis, in ECCV 2022.

5. I am very curious that the FGD and FVD in Experimental Table 1 seem a bit weird. While the authors' method achieves a larger marginal improvement in FGD than the suboptimal S2G (1.303 vs 23.646). The FVG of these two methods is very propinquity (476.120 vs 486.134). Does this mean that the author's proposed ``STRUCTURE-AWARE IMAGE REFINEMENT'' is not effective?

6. The proposed ``learnable edge heatmaps'' is similar to the work named [4]. I suggest the authors compare with it.

[4] Audiodriven neural gesture reenactment with video motion graphs, in CVPR 2022.

**Questions:**

Please refer to Weaknesses

---

> ### Author Response · Authors · 2024-11-18
>
> ### Reply to Paper Flow Delivery Issue:
> Thank you for your constructive feedback. In response, we have revised the manuscript to address your concern by reducing the space dedicated to related works in the introduction and refocusing the section to more clearly highlight the technical contributions of our work. We hope that this revision better emphasizes the motivation and key insights behind our approach. Please refer to the updated manuscript for further details.
>
> ---
>
> ### Reply to Writing Issue:
> We appreciate your careful review and have made the necessary revisions to the manuscript. Specifically, we have included the audio transcript and corrected the typo. We trust that these changes improve both the clarity and readability of the text.
>
> ---
>
> ### Reply to Similar Method [4]:
> Thank you for your suggestion. While we acknowledge the similarity with [4], we believe our work offers a significant distinction. The graph in [4] focuses on a subsequence of motion, while our method leverages an edge heatmap that captures the connectivity of joints within a single pose. Moreover, due to the closed-source nature of [4], we were unable to reimplement it for a direct comparison. We hope this clarification resolves the concern.
>
> ---
>
> ### Reply to Technical Novelty Concern:
> Thank you for raising the concern regarding the technical novelty of our approach. We understand that **VQ + Masking-representation Learning** is not new across various domains, but we would like to emphasize the unique contributions of our work.
> In contrast to previous works like [1], [2], and [3], which rely on conventional VQ-based motion tokens, we introduce **Contextualized Motion Representation** as the major innovation. It lies in **contextual distillation via contrastive alignment for learning speech triggers** for gesture patterns (as described in Section 4.1). To our knowledge, this is the first attempt to explore the benefits of **audio-motion representation learning** for constructing **contextualized motion representations for gesture generation**. Our approach directly integrates alignment information into the motion representation using an **RQ-codebook**, which allows us to fuse contextual triggers from speech (such as pronouns like "this" or "they") into the motion embedding. This enables the generator to accurately map speech triggers to corresponding motion representations during generation.
>
> Further distinguishing our approach, we propose an **Iterative Re-masking-Based Generator Design**, which is optimized for **co-speech gesture generation**. Unlike autoregressive or diffusion models, which are often slow and not suitable for real-time generation of long sequences, our method can generate gestures in just **5 steps**. We demonstrate that additional steps beyond this can actually degrade quality by disrupting the temporal coherence between modalities (a challenge that does not arise in text-to-motion domains).
>
> Our extensive experiments, particularly those in Table 3 (b) and (c), demonstrate that applying modality alignment alone yields an FGD score of around 21. However, incorporating contextual distillation through contrastive learning significantly improves performance, resulting in better contextual awareness during generation.
>
> We believe our **generator design and iterative re-masking approach** present significant advantages in terms of both speed and quality, and the experiments in Tab. (e) and (f) in the main paper clearly highlight these benefits.
>
> To provide further clarity and quantitative comparisons, we include additional experiments and analysis based on a single speaker (Scott) in the revised supplementary material, along with a response to Reviewer LNUW. Demo videos showcasing our approach are also available [here](https://anonymousaisubmission.github.io/) and is included in the updated supplementary material. We also provide a quantitative comparison of long-sequence generation on BEAT-X in the supplementary materials, which we believe underscores the advantages of our method.
>
> | Method                         | FGD (↓) | BC (↑)  | Diversity (↑) | MSE (↓)    | LVD (↓)   |
> |---------------------------------|---------|---------|---------------|------------|-----------|
> | Rhythmic Gesticulator [AO et al. 2022] | 6.453   | 6.558   | 9.132         | -          | -         |
> | TalkSHOW [Yi et al. 2022]       | 6.209   | 6.947   | 13.47         | 7.791      | 7.771     |
> | EMAGE [Liu et al. 2023]         | 5.512   | **7.724**| 13.06         | 7.680      | 7.556     |
> | Ours (w/o Distillation)         | 7.479   | 7.395   | 12.12         | 7.656      | 7.671     |
> | **Ours**                        | **4.650**| 7.370   | **13.55**     | **7.343**  | **7.432** |

---

> > ### Author Response · Authors · 2024-11-23
> >
> > ### Reply to Metric Comparison:
> > Thank you for your thoughtful comments on our metric comparison. We acknowledge that the FVD score shows a smaller difference (476.120 vs. 486.134) compared to the FGD score (1.303 vs. 23.646). However, we believe this is due to the different sensitivities of the two metrics. FVD is primarily focused on pixel-level differences, which do not fully capture the quality of motion patterns or keypoint locations. On the other hand, FGD is more sensitive to these factors, providing a more strict measure of the naturalness of the generated motions.
> >
> > In our ablation study (Tab. (d)), we show that without image refinement, the FVD score is **492.341**, but when image refinement is applied, it drops to **476.120**, highlighting the effectiveness of our **Structure-Aware Image Refinement**. Furthermore, other visual quality metrics, such as VQA$_T$ and VQA$_A$, also reflect significant improvements in image quality due to this module. For visual comparisons, our supplementary videos provide clear evidence of the substantial quality difference, with our method rendering more natural and coherent hand movements and reducing issues like blurry shoulders and unnatural motion.
> >
> > We hope these clarifications address your concerns and demonstrate the strength of our approach.
> >
> > ---
> >
> > We appreciate your thoughtful feedback and hope that our revised manuscript, along with the additional clarifications, more clearly communicates the novelty and effectiveness of our approach.

---

> ### Comment · Reviewer_P3xu · 2024-11-25
> **Response to authors**
>
> Dear authors,
>
> Thanks for your efforts to address my concerns.
>
> I have read the comments of all the reviewers. While I still have some concerns about the metrics in FGD and FVD, I will maintain my score of 5 in the final rating.
>
> Best
>
> Reviewer P3xu

---

### Official Review · Reviewer_bDof · 2024-11-03

**Soundness:** 2
**Presentation:** 2
**Contribution:** 2
**Rating:** 3
**Confidence:** 4

**Summary:**

The paper addresses challenges in co-speech gesture video generation through a proposed method called Realistic-Gesture. This method incorporates three main components: a speech-aware gesture motion representation, a masked gesture motion generator, and a pixel-level refinement module. Experimental results show that the proposed approach generates realistic co-speech gesture videos, while also enabling long-sequence generation and video editing capabilities.

**Strengths:**

1. The authors identify three key challenges in co-speech gesture video generation and propose solutions to address each one.

2. Amount of ablation studies are conducted to verify the effectiveness of the proposed modules.

**Weaknesses:**

Generally, some challenges identified in this paper have been addressed in prior research; however, the authors failed to cite these studies or compare their findings with the experiments conducted in this work. Notable examples include:

1. **"Rhythmic Gesticulator: Rhythm-Aware Co-Speech Gesture Synthesis with Hierarchical Neural Embeddings."** This study proposed a shared hierarchical embedding for both speech content and motion, which closely resembles the approach taken by the authors.

2. **"Make-Your-Anchor: A Diffusion-based 2D Avatar Generation Framework."** This research adopted SMPL-X parameters as motion representations for co-speech video generation, enhancing the clarity of hand movements.

Additional weaknesses include:

1. While the proposed method can perform gesture inpainting, the authors inaccurately claim that it supports video gesture editing. This assertion is misleading, as the inpainted gestures are not controllable.

2. In some of the "Gesture Pattern Transfer" videos, the character appears distorted, likely due to differences in body proportions between the source and target characters.

**Questions:**

See weaknesses part.

---

> ### Author Response · Authors · 2024-11-18
>
> ### Reply to Missing Citations and Comparisons:
> Thank you for pointing out the missing related works. We have now included two relevant studies in the related work section: one on gesture generation and the other in conditional video generation. **While both are relevant works for gesture and video generation separately, neither provides a unified framework like ours.** We apologize for initially overlooking them and have added references to these works in the new revision (Line 102 and Line 120, respectively).
>
> For **gesture generation comparison**, we have included the additional experiments in the revised supplementary videos and defer more about the detailed comparions to gesture-generation-only models including **Rhythmic Gesticulator** in response to Reviewer LNUW and Reviewer P3xu.
>
> Regarding **video synthesis**, the **Make-Your-Anchor** (Stable-Diffusion-based methods) requires **complex** preprocessing with 3D tracking to obtain SMPL-X, and **does not provide fast and computational-friendly way** as we do for video avatar animation. In addition, without the released training code or data preprocessing details before ICLR submission and right now, we were unable to replicate it. Therefore, due to its similarity with **AnimateAnyone**, we have opted to use it as a replacement and defer the time and resource comparison, in response to Reviewer XVKk. For visual synthesis quality comparison, we provide **addition expriments in the supplementary material and demo videos**. Diffusion-based models **lack of background motion control** and present significantly background temporal jittering problem.
>
> ---
>
> ### Reply to Gesture Editing:
> We acknowledge that the explanation of gesture editing was insufficient and may have caused confusion. To clarify, the gesture editing process includes two main types:
>
> 1. Editing the intermediate segment of a video to transfer the style of one speaker to another (as shown in the original demo).
> 2. Replacing a small portion of audio in the original video with new audio and rerendering the video. (as shown in the revised demo video)
>
> ---
>
> ### Reply to Distorted Gesture Pattern Transfer:
> Thank you for pointing out the distortion in some of the "Gesture Pattern Transfer" videos. We recognize that these issues arise due to mismatched body proportions between source and target characters, and are limited by the available data on speaker identities. We hope that future work of scaling up the training will help address this issue.
>
> ---
>
> We hope our explanations can provide clear presentations of our work and address your concern.

---

> > ### Comment · Reviewer_bDof · 2024-11-29
> > **Response to authors**
> >
> > Dear Authors,
> >
> > Thank you for your response and the additional experiments. I have also reviewed the comments provided by the other reviewers. After carefully considering your response, I still have concerns regarding the following aspects:
> >
> > 1. While the authors claim to present a unified framework for co-speech gesture video synthesis distinct from prior methods, the use of 2D keypoints as intermediate representations between the co-speech gesture synthesis module and the video synthesis module does not appear to be a novel approach.
> >
> > 2. The gesture editing capabilities described by the authors can be achieved by combining any co-speech gesture synthesis model with a co-speech gesture video generation model. This suggests that these capabilities are not unique to the proposed method.
> >
> > Overall, this work enhances the co-speech gesture synthesis module and the gesture video synthesis module. However, as noted by Reviewer P3xu and Reviewer zy2t, the core components of these modules closely resemble those in previous studies. Therefore, I will maintain my initial score for the final rating.

---

### Author Response · Authors · 2024-11-20

We would like to thank all reviewers for their invaluable comments and feedback. We have replied to each reviewer individually to address their questions and concerns, and have updated the paper to include extra results, figures, and demo videos. We would like to highlight and answer some concerns raised in common:

---

### Novelty and Contribution Explanation
To present the contribution of this work over existing literature, **1)** We are the first work to explore benfits of audio-motion representation learning for constructing contextualized motion representations for gesture generation **2)** We do extensive experiments to validate the masking based model design with fast inference speed of **5** steps, with various downstream application support; **3)** We propose a structure-ware image refinement method to resolve the ambiguity by image-warping for pixel-level avatar animation. With these three, we are capably of achieving a joint co-speech gesture motion generation and photo-realistic avatar video animation within one unified framework.

---

### Comparison with gesture generation only methods
We have included the comparison between our method with **EMAGE**, **TalkShow**, and **Rhythmic Gesticulator**, in the new supplementary document Tab.2 and corresponding demo videos beat-x-comparison.mp4 in the revised demo page . It shows better performance over existing SOTA methods. We in addition present the ablation videos ab-1.mp4, ab-2.mp4, ab-3.mp4 for PATS dataset and also beat-x-comparion2.mp4 and beat-x-comparion3.mp4 for BEAT-X dataset, demonstrating the benefits of contextual alignment and distillation for motion representation learning's benefits in our framework.

---

### Comparison with avatar video generation only method
As suggested by Reviewer bDof, we also introduces the visual comparions with AnimateAnyone, which indicate the Diffusion-based model lacks of capability of disentangle camera motion and human motions in the videos, leading to significant background temporal jittering problem.

 ---

We hope our answers and updated paper can help to address the concerns raised in the initial reviews. Since we received highly mixed comments from the reviewers, we would highly appreciate active discussions from the reviewers and we are happy to clarify any further questions.

---

### Meta-Review · Area_Chair_NPQQ · 2024-12-19

**Metareview:**

This paper received mixed reviews and received an overall borderline rating. Reviewers have pointed out the issues of the generated videos in the supplementary materials, and the core components are not sufficiently novel. AC also checked the generated videos, and agreed with the reviewers that the generated video quality is subpar. For instance, the hands are quite blurry in the generated results of the supplementary materials. Reviewers P3xu, zy2t and bDof, all raised the concerns that the core components of these modules closely resemble previous studies. Based on the inferior generation results and insufficient novelty, AC made the final decision.

**Additional Comments On Reviewer Discussion:**

The most and critical issues for this work are the generation quality and novelty of the proposed components. Reviewers P3xu, zy2t and bDof all raised the similar concerns in the initial reviews. The authors provided feedback on the raised concerns and some concerns of the reviewers have been effectively addressed. However, reviewer bDof pointed out two works which are highly related to the proposed components and shared concerns among reviewers. In addressing the novelty, the authors emphasize the novelty lies in the unified framework, but the final results are not superior. AC agreed with reviewers that the long-sequence generation quality is indeed not superior. Therefore, the contributions of the proposed components seem not significant.

---

### Decision · Program_Chairs · 2025-01-22

Reject